# Longitudinal Event-Level Sexual Risk and Substance Use among Gay, Bisexual, and Other Men Who Have Sex with Men

**DOI:** 10.3390/ijerph18063183

**Published:** 2021-03-19

**Authors:** Jordan M. Sang, Zishan Cui, Paul Sereda, Heather L. Armstrong, Gbolahan Olarewaju, Allan Lal, Kiffer G. Card, Eric A. Roth, Robert S. Hogg, David M. Moore, Nathan J. Lachowsky

**Affiliations:** 1British Columbia Centre for Excellence in HIV/AIDS, Vancouver, BC V6Z 1Y6, Canada; cuizishan@gmail.com (Z.C.); psereda@bccfe.ca (P.S.); golarewa@gmail.com (G.O.); alal@bccfe.ca (A.L.); or robert_hogg@sfu.ca (R.S.H.); dmoore@bccfe.ca (D.M.M.); nlachowsky@uvic.ca (N.J.L.); 2School of Psychology, University of Southhampton, Southampton SO17 1BJ, UK; H.Armstrong@soton.ac.uk; 3School of Public Health and Social Policy, University of Victoria, Victoria, BC V8P 5C2, Canada; kiffercard@gmail.com (K.G.C.); ericroth@uvic.ca (E.A.R.); 4Canadian Institute for Substance Use Research, University of Victoria, Victoria, BC V8P 5C2, Canada; 5Faculty of Health Sciences, Simon Fraser University, Burnaby, BC V5A 1S6, Canada; 6Department of Medicine, Faculty of Medicine, University of British Columbia, Vancouver, BC V6T 1Z4, Canada

**Keywords:** sexual risk, trends, event-level, substance use, MSM, STI

## Abstract

(1) Background: Condomless anal sex and substance use are associated with STI risk among gay, bisexual, and other men who have sex with men (gbMSM). Our first study objective was to describe event-level sexual risk and substance use trends among gbMSM. Our second study objective was to describe substances associated with event-level sexual risk. (2) Methods: Data come from the Momentum Health Study in Vancouver, British Columbia and participants were recruited from 2012–2015, with follow-up until 2018. Stratified by self-reported HIV status, we used generalized estimating equations to assess trends of sexual event-level substance use and assessed interactions between substance use and time period on event-level higher risk sex defined as condomless anal sex with an HIV serodifferent or unknown status partner. (3) Results: Event-level higher risk anal sex increased across the study period among HIV-negative/unknown (baseline prevalence: 13% vs. study end prevalence: 29%) and HIV-positive gbMSM (baseline prevalence: 16% vs. study end prevalence: 38%). Among HIV-negative/unknown gbMSM, event-level erectile drug use increased, while alcohol use decreased over the study period. Overall, interactions between substance use and time on higher risk anal sex were not statistically significant, regardless of serostatus. However, we found a number of time-specific significant interactions for erectile drugs, poppers, Gamma-hydroxybutyrate (GHB), crystal methamphetamine and ecstasy/MDMA use among HIV-negative/unknown gbMSM. (4) Conclusion: Significant differences in substance use trends and associated risks exist and are varied among gbMSM by serostatus. These findings provide a more comprehensive understanding of the effects of event-level substance use on sexual risk through longitudinal follow-up of nearly six years.

## 1. Introduction

Rates of sexually transmitted infections (STI) are rising in Canada and are highly concentrated among gay, bisexual, and other men who have sex with men (gbMSM) [1]. Further, disparities in STI prevalence, including diagnoses of chlamydia, gonorrhea, syphilis, and lymphogranuloma venereum (LGV), suggest HIV serostatus plays an important role in STI transmission [1,2]. Serosorting is a common practice among gbMSM, where individuals select sexual partners based on HIV status. However, serosorting based on HIV status may increase rates of STIs as gbMSM may forgo condoms with the same HIV serostatus partners, concentrating STI rates among HIV-positive gbMSM [3]. Moreover, higher rates of serodiscordant condomless anal sex (CAS) are reported among HIV-positive gbMSM compared to HIV-negative gbMSM [4], with serodiscordant CAS identified as the main factor in STI diagnoses among HIV-positive among gbMSM [5]. Within Canada, a study of gbMSM living with HIV found an increased risk of chlamydia and gonorrhea was associated with multiple HIV-positive partners as well as recreational drug use [6]. Collectively, STI’s may be more prevalent among certain sexual networks of gbMSM and considerations of HIV serostatus on STI risk are important to further understand sexual risk.

Substance use is a significant factor in influencing sexual risk-taking behaviors, such as CAS. Previous literature on substance use patterns among gbMSM indicates variability in different substance types and distinct classes of substances, such as club drug use, sex drug use and conventional drug use [7]. A systematic review on sexualized drug use among gbMSM found multiple factors that may promote engagement in sexualized drug use, such as coping with stressful events, increasing intimacy, fulfilling community belonging and enhancing sexual performance and functioning [8]. Related, chemsex refers to certain recreational drugs (particularly combinations of crystal methamphetamine, mephedrone and gamma-hydroxybutyrate/gamma-butyrolactone (GHB/GBL)) used before or during sex, which help facilitate or enhance sex [9]. Among gbMSM living with HIV, multiple studies have found chemsex significantly associated with serodiscordant CAS, serodiscordant CAS with a partner, who has a detectable viral load, increased number of sexual partners, and increased STI diagnoses [10,11]. Among HIV-negative gbMSM, existing literature found chemsex was associated with serodiscordant/HIV-unknown CAS, STI diagnoses, a greater number of sexual partners and group sex events [12]. Mixed-method research from the United Kingdom help contextualize findings as gbMSM reports difficulty negotiating safer sex while under the influence of substances, and perceptions about HIV and STI sexual risk may also be skewed [13].

The association between substance use and sexual risk relates to situational events, which are highly contextual and may vary in terms of substances used, partner characteristics, condom usage, and sexual acts. To address these limitations, various event-level approaches focusing on specific sexual encounters and substance use within highly restricted time periods before or during sex may provide nuanced understanding [14]. A review of the literature on event-level substance use and sexual behaviors among gbMSM found consistent associations of sexual risk with methamphetamine use and alcohol binge drinking among gbMSM [15]. However, inconsistencies in event-level measurement and analysis may limit generalizability in findings. For example, using both retrospective and prospective event-level data, Rendina et al. (2015) found event-level substance use, in particular, club drugs, such as ketamine, ecstasy/3,4-methylenedioxymethamphetamine (MDMA), GHB, cocaine/crack, or methamphetamine, increased odds of sex and engaging in CAS [16]. Among gbMSM living with HIV, Sullivan and colleagues found self-reported heavier alcohol users reported less frequent condom use [17]. Associations between substance use and CAS are also not limited by age, as consistent event-level diary findings among young gbMSM also found associations between alcohol use and CAS with casual partners [18]. While event-level substance use has been associated with CAS among gbMSM, overall event-level substance use has not been found to be associated with perceived control or pleasure during sex [19]. However, individual substances such as crystal methamphetamine use has been found to be significant for both CAS and lower perceived control during sex [19]. Taken together, various event-level approaches highlight evidence for associations between substance use and sexual risk. However, research within a Canadian context is limited.

Using event-level longitudinal data to measure STI sexual risk, our first study objective was to describe event-level sexual risk and substance use trends in Vancouver during treatment as prevention scale-up (TasP) over a six-year period [20]. Our second objective was to describe individual and partner substance use associated with event-level sexual risk and to determine whether there have been any changes in the significance of certain substances as they relate to condom use over the study period.

## 2. Materials and Methods

### 2.1. Study Protocol and Participants

Data are from the Momentum Health Study, a prospective longitudinal, bio-behavioral study of gbMSM in Vancouver, British Columbia. Participants were recruited from February 2012 to February 2015 using respondent-driven sampling (RDS). RDS recruitment involved initial “seed” participants, who were recruited through community partner agencies and online advertisements on gbMSM social networking websites and apps. The full RDS methodology of our study has been published elsewhere [21]. To be eligible, participants had to gender-identify as a man, be 16 years of age or older, report having sex with another man in the past six months, currently live in Metro Vancouver, and be able to complete the questionnaire in English. Participants completed a 90 min in-person study visit every six months, which included a computer-assisted self-interview (CASI) and study nurse visit. Participants received a $50 CAD honorarium for their participation and could receive an additional $10 CAD for each eligible participant they referred that completed the study (maximum of six). Study visits up to February 2018 are included in this analysis. All participants signed an Informed Consent form about the study and their involvement. The research protocol and human ethics clearances were approved by The University of British Columbia, Simon Fraser University, and The University of Victoria.

### 2.2. Outcome Variable

The primary outcome variable was higher risk anal sex, which was defined as any CAS with an HIV serodifferent or unknown status partner. Participants were asked to complete a “partner matrix” of a repeating set of questions about their last sexual encounter with each of up to five of their most recent sexual partners within the past six months (maximum of five partners). Sexual encounters that did not include anal sex were excluded in this analysis. Condom use was reported for each partner, and participants indicated their use/non-use as the receptive and insertive partner. Any form of CAS (receptive or insertive) was included in this analysis. The partner’s HIV serostatus was obtained by asking participants if they knew their partner(s)’s HIV status before having sex, what the partner’s status was, and how they knew their partner’s status (if they knew). From these, we determined if the partner’s serostatus was positive, negative, or unknown.

### 2.3. Explanatory Variables

The primary explanatory factors were time (for trend analyses) and substance use. Time of event was assessed with a six-month period prevalence between study visits over the course of almost six years. Event-level factors were collected for each partner and reported sexual event. Participants indicated the number of male sexual partners, the number of months since they first had sex with each partner, and the number of times they had anal sex with each partner in the past six months (per act). Participants indicated the month and year of the last sexual event with each partner, which was used to conduct a change over time analysis. For each sexual event, participants indicated their anal sex positions (receptive, insertive, or both), their level of certainty regarding their partner’s HIV status before sex, whether they expected they would have sex with this partner again, and whether they received any goods, money, drugs, or services in return for sex. Participants reported their own and their partner’s substance use in the two hours prior to and during each sexual event, which included any alcohol, cannabis, erectile drugs, poppers (amyl nitrate), crystal methamphetamine, GHB, and MDMA.

Psychosocial variables included the HIV treatment optimism-skepticism scale (12 questions, range: 12–48, study α = 0.85) [22], the 11-item sexual seeking scale (revised) (range: 11–44, study α = 0.73) [23], the 7-item personal (range: 1–5, α = 0.75) and 6-item communal subscales for the sexual altruism scale (range: 1–5, study α = 0.77) [24], and the 10-item alcohol use disorders identification test (AUDIT) (range: 0–40, study α = 0.86) [25].

Demographic variables included participants’ age, sexual orientation, race/ethnicity, annual income, education, residence, and relationship status. We also asked a series of potential HIV prevention or risk-reduction practices (i.e., always using condoms, seropositioning, serosorting, viral-load sorting, abstinence, withdrawal, asking for HIV status before sex), PrEP usage, escort work and attending group sex events in the past six months. For HIV-positive gbMSM, we utilized the study linkage to the BC Centre for Excellence in HIV/AIDS’s Drug Treatment Program administrative database to assess treatment adherence and viral load [20].

### 2.4. Analysis

We limited our analyses to the sexual-event level and stratified participants by self-reported HIV status. Generalized estimating equations (GEE) were used to construct hierarchical logistic regression models, adjusting for participant interdependence in the data (events within participants as main clusters and each visit as sub-clusters). We examined trends over time with higher risk anal sex and substance use. Furthermore, we also tested interactions for substances and time to assess whether their associations with higher risk anal sex significantly changed over the study period. Odds ratio per six-months are presented, and significance was assessed as a *p-*value *<*0.05. RDS weighting was not applied, given that the analysis is based on event-level data. We included post hoc lost-to-follow-up analyses (LTFU) in determining significant differences between participants who did not complete the study and our final sample. All analyses were conducted using SAS version 9.4 (SAS, Cary, NC, USA).

## 3. Results

### 3.1. Descriptive Results

The median follow-up time for participants was 3.03 years. 549 HIV-negative/unknown gbMSM reported 8121 anal sexual events, of which 17.9% included CAS with a serodifferent or unknown status partner. Among the 213 HIV-positive gbMSM at baseline, 3454 anal sexual events were reported, of which 27.9% included CAS with a serodifferent or unknown status partner. Full descriptive statistics on the sample stratified by HIV status can be found in Table 1.

### 3.2. Analytical Results

In our GEE, higher risk anal sex was significantly associated with greater use of poppers, erectile drug use, ecstasy/MDMA use, GHB use, and crystal methamphetamine use among HIV-negative/unknown gbMSM; only poppers use was significantly associated with greater odds of higher risk anal sex for HIV-positive gbMSM. Full results can be found in Table 2.

For trends among HIV-negative/unknown gbMSM, we found that higher risk anal sex events increased over time (first time period prevalence: 13%, last time period prevalence: 29%) (OR = 1.006; 95% CI = 1.002, 1.011, *p* = 0.009). We found event-level CAS increased over the study period (OR = 1.015; 95% CI = 1.011, 1.019, *p* < 0.001), anal sex with HIV-negative partners increased over the study period (OR = 1.013; 95% CI = 1.009, 1.017, *p* < 0.001), and anal sex with HIV-positive gbMSM increased over the study period (OR = 1.012; 95% CI = 1.006, 1.017, *p* = 0.000). In relation to event-level substance use among HIV-negative/unknown gbMSM over the study period, event-level alcohol use decreased (first time period prevalence: 46%, last time period prevalence: 27%) (OR = 0.989; 95% CI = 0.985, 0.993, *p* < 0.001), and erectile drug use increased (first time period prevalence: 6%, last time period prevalence: 9%) (OR = 1.010; 95% CI = 1.001, 1.020, *p* = 0.031). Full results can be found in Figure 1.

For trends among HIV-positive gbMSM, our model found higher risk anal sex increased over time (first time period prevalence: 16%, last time period prevalence: 38%) (OR = 1.006; 95% CI = 1.001, 1.012, *p* = 0.025). In relation to event-level substance use, we found that popper use decreased over time (first time period prevalence: 38%, last time period prevalence: 31%) (OR = 0.991; 95% CI = 0.984, 0.998, *p* = 0.007). Full results can be found in Figure 2.

We did not find any significant interactions between time and substance use on the likelihood of event-level higher-risk anal sex. However, we did find a number of time-specific significant interactions. Full results can be found in Table 3.

We conducted a post hoc lost to follow-up analysis and found that participants LTFU reported less higher-risk anal sex among HIV-negative/unknown gbMSM and more ecstasy/MDMA use among HIV-positive gbMSM.

## 4. Discussion

Our research explored temporal trends and associations between substance use and higher risk sex during anal sex events among gbMSM in Vancouver, BC. We found that event-level sexual risk increased over time for both HIV-negative/unknown gbMSM and HIV-positive gbMSM over nearly six years of follow-up. However, HIV-negative/unknown gbMSM reported more frequent CAS with serodifferent or unknown status partners compared to HIV-positive gbMSM. Although we did not find longitudinally significant interactions between substance use and time on higher-risk anal sex, we found a number of time-specific associations that warrant further exploration.

We found event-level higher risk anal sex increased over time for both HIV-negative/unknown gbMSM and HIV-positive gbMSM. Among HIV-negative/unknown gbMSM, we also found increasing trends of overall CAS. Trends of gbMSM gaining regular partnerships and fewer sex partners may better explain this finding [26]. As gbMSM are more likely to have a regular partner, they may also be less likely to wear condoms with their regular partner over time [27]. Additionally, we found increasing trends of CAS with known HIV-negative versus unknown partners and CAS with known HIV-positive versus unknown partners. We hypothesize that based on increasing trends of HIV testing across Canada [28], gbMSM are also reporting fewer unknown HIV status partners over time.

Exploring substance use trends among HIV-negative/unknown gbMSM, we found alcohol use decreased and erectile drugs increased over time. The cohort aging effect may explain decreases in alcohol as findings indicate alcohol consumption and use of illicit substances tend to decrease across the lifespan [29]. In relation to erectile drug use, evidence suggests erectile drugs are commonly used by gbMSM, who are more sexually active and who use erectile drugs specifically to enhance sexual performance and duration [30]. The longitudinal nature and aging cohort effect may further explain increases in erectile drug use over time. However, we did not find these trends among HIV-positive gbMSM. Instead, we found popper use decreased over time among HIV-positive gbMSM. This finding is significant because, in 2013, Canada banned the sale of poppers, which may have precipitated this decline. Although the sale of poppers has been banned, critics argue that this ban is not supported by science and that this has led to an unregulated market, increasing the risk of dangerous off-market products and forcing access to poppers in potentially dangerous ways [31]. A recent analysis on popper use among young gbMSM found high lifetime and recent popper use, yet dependency symptoms and risky consumption or problems arising from using poppers were low [32]. Overall, differences in substance use trends among gbMSM by serostatus have been identified elsewhere, yet further research is needed to delineate unique event-level substance use factors by serostatus [33].

We identified noteworthy findings from our univariable analyses. First, our findings for PrEP usage should be interpreted with caution given the limited number of gbMSM reporting ever using PrEP, which was less than 2% of participants reporting using PrEP at any point in our study. Recently, in British Columbia, PrEP became publicly funded and freely available in January 2018 [34]. Thus, our analysis provides a basis for further exploration of PrEP use, substance use and sexual risk as PrEP use increases. It is important to note that biomedical interventions, such as PrEP, are changing gbMSM’s notions of “safe” and “risky” sex, as CAS with a serodiscordant or unknown partner may not place increased HIV risk to individuals on PrEP. However, also important to consider is that biomedical interventions only protect against HIV, and STI risk may still be associated with CAS. Thus, condoms still provide relevant protection and should be paired with health promotion and programming that embraces a broader STBBI framework. Second, we did not find a significant difference between viral-load status and higher risk anal sex, which is inconsistent with previous findings [35]. We hypothesize that these differences may be explained by the low proportion of gbMSM with a detectable viral load in our study. At baseline, only 16% of HIV-positive gbMSM reported a viral load of ≥200, and this rate is expected to have gone down as TasP scale-up increased. Third, our results found greater endorsement of HIV treatment optimism was associated with higher odds of sexual risk. Previous research exploring HIV treatment optimism among gbMSM found increasing trends of HIV-optimism over time, but no longitudinal differences in higher risk anal sex for both HIV-negative and HIV-positive men [36]. Differences may be explained by our focus on event-level sexual risk and variations in sexual behaviors with specific partners versus overall sexual risk behaviors.

We did not find statistically significant interactions between time and substance use on higher-risk anal sex. However, we found that erectile drug use and poppers use were consistently associated with higher risk anal sex throughout the study period for HIV-negative/unknown gbMSM. These findings are consistent with previous literature, which demonstrates individual associations between erectile drug use and popper use on sexual risk [37,38]. Interestingly, we found GHB, crystal methamphetamine, and ecstasy/MDMA use associated with sexual risk at one point in our study (baseline or end of study), but the associations were not maintained throughout the study period. We hypothesize that interactions between substance use and sexual risk over time were not significant because rates of substances may have varied over time with different partners. As such, gbMSM, who engage in multiple sex partners, may have had significant sexual risk at one point in our study, but sexual partners and substances used may have changed over time. Alternatively, gbMSM may have had fewer sexual partners over time, whereby sexual risk and substance use decreased with time and partnership.

This research is subject to limitations. First, although our research used sexual event-level data, these are likely not representative of all sexual events between partners and are subject to recall bias. Still, our focus on event-level data is advantageous in comparison to cross-sectional data to causally examine the relationship between substance use and sexual risk. Second, our findings may not be comparable outside of Vancouver, British Columbia, especially given the TasP context and promotion in the city. Third, we did not distinguish between different partner types (e.g., casual or main partners), which may influence the use of substances, sexual behaviors, and the relationship between these. Fourth, we could not distinguish between medically prescribed erectile drugs and non-medically prescribed. However, the prevalence of recreational erectile drug use among gbMSM is high, and erectile drug use (regardless of the prescription) has been identified as an important indicator of sexual risk in existing literature [39,40]. Fifth, many of our significant findings for substance use and higher risk anal sex diminished over time. We theorized this might be due to the LTFU of high-risk/high substance-using participants and completed post hoc analyses to assess this. We found two significant differences for participants LTFU, and these findings may limit the scope of interpretation for these results.

## 5. Conclusions

Our findings indicate that sexual risk is associated with event-level substance use, behavioral and psychosocial factors. These findings provide a more comprehensive understanding of the effects of event-level substance use on sexual risk among Canadian gbMSM through longitudinal follow-up. Moreover, future interventions must address increasing rates of sexual risk through continued education on STIs, increased access to safer sex materials, and continued preventative screening. Understanding the nuances of individual substances as well as polysubstance use will be beneficial in developing targeted interventions for substance-using gbMSM in managing sexual risk. By focusing on separate sexualized events with different partners and substances used, sexual risk programs can target strategies for reducing risk in certain scenarios and how individuals can apply this knowledge to future sexual events.

## Figures and Tables

**Figure 1 ijerph-18-03183-f001:**
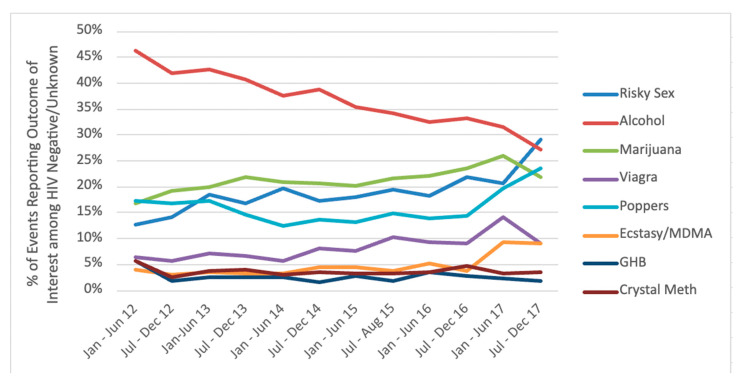
Percentage trends of sexual events reporting substance use and higher risk anal sex among HIV-negative/unknown gbMSM in Metro Vancouver. Notes: MDMA = 3,4-methylenedioxymethamphetamine; GHB = 3,4-methylenedioxymethamphetamine.

**Figure 2 ijerph-18-03183-f002:**
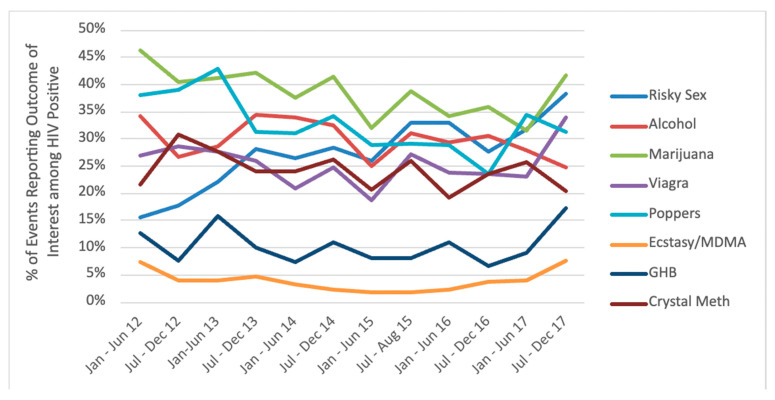
Percentage trends of sexual events reporting substance use and higher risk anal sex among HIV-positive gbMSM in Metro Vancouver. Notes: MDMA = 3,4-methylenedioxymethamphetamine; GHB = 3,4-methylenedioxymethamphetamine.

**Table 1 ijerph-18-03183-t001:** Baseline participant-level descriptors of gbMSM in Metro Vancouver, stratified by self-reported HIV status.

Variables	Overall		HIV-Negative/Unknown		HIV-Positive	
	*N*		*n*	%	*n*	%
	762		549	72	213	28
Age						
16–29	288	37.8	277	50.5	11	5.2
30–39	254	33.3	180	32.8	74	34.7
40+	220	28.9	92	16.8	128	60.1
Sexual Orientation						
Gay	647	84.9	464	84.5	183	85.9
Bisexual	69	9.1	50	9.1	19	8.9
Other	46	6.0	35	6.4	11	5.2
Ethnicity						
White	577	75.7	408	74.3	169	79.3
Asian	74	9.7	62	11.3	12	5.6
Indigenous	46	6.0	28	5.1	18	8.5
Latino	35	4.6	28	5.1	7	3.3
Other	30	3.9	23	4.2	7	3.3
Born in Canada						
No	176	23.1	140	25.5	36	16.9
Yes	586	76.9	409	74.5	177	83.1
Neighborhood						
Downtown/West End	375	49.2	234	42.6	141	66.2
Elsewhere Vancouver	238	31.2	192	35.0	46	21.6
Outside Vancouver	149	19.6	123	22.4	26	12.2
Formal Education						
Some or completed high school	170	22.3	109	19.9	61	28.6
Any post-secondary training	592	77.7	440	80.2	152	71.4
Annual Income						
<$30,000	476	62.5	324	59.0	152	71.4
at least $30,000	286	37.5	225	41.0	61	28.6
Current Regular Partner						
No	470	61.7	337	61.4	133	62.4
Yes, but not common law/married	163	21.4	120	21.9	43	20.2
Yes, common law/married	129	16.9	92	16.8	37	17.4
Usage of PrEP						
No	141	23.7	85	19.0	56	38.1
Yes	1	0.2	0	0.0	1	0.7
Never heard of PrEP	453	76.1	363	81.0	90	61.2
Escort Work						
No	625	82.0	481	87.6	144	67.6
Yes, in P6M	47	6.2	30	5.5	17	8.0
Yes, not in P6M	90	11.8	38	6.9	52	24.4
Attended Group Sex P6M						
No	565	74.2	427	77.8	138	64.8
Yes	197	25.9	122	22.2	75	35.2
AUDIT Zone						
Low-risk (scores 0 to 7)	455	60.0	293	53.5	162	76.8
Medium-risk (scores 8 to 15)	203	26.8	170	31.0	33	15.6
Harmful (scores 16 to 19)	54	7.1	45	8.2	9	4.3
Possible dependence (scores 20 and over)	47	6.2	40	7.3	7	3.3
On ART Ever						
No	3	1.4			3	1.4
Yes	213	98.6			210	98.6
Treatment Adherence P12M						
95% or greater	120	55.6			120	56.3
<95%	53	24.5			53	24.9
Never on ART/start within 12 M	43	19.9			40	18.8
Latest Viral Load <200 copies/mL						
No	37	17.1			34	16.0
Yes	179	82.9			179	84.0
Prevention Strategies						
Always Using Condoms						
No	334	44.1	190	34.9	144	67.6
Yes	424	55.9	355	65.1	69	32.4
Seropositioning						
No	539	71.1	400	73.4	139	65.3
Yes	219	28.9	145	26.6	74	34.7
No Anal Sex						
No	411	54.2	275	50.5	136	63.9
Yes	347	45.8	270	49.5	77	36.2
Serosorting						
No	453	59.8	352	64.6	101	47.4
Yes	305	40.2	193	35.4	112	52.6
Viral-Load Sorting						
No	617	81.4	490	89.9	127	59.6
Yes	141	18.6	55	10.1	86	40.4
Withdrawal						
No	538	71.0	392	71.9	146	68.5
Yes	220	29.0	153	28.1	67	31.5
Asking Status						
No	314	41.4	211	38.7	103	48.4
Yes	444	58.6	334	61.3	110	51.6
Event-Level Outcomes	*n*	%	*n*	%	*n*	%
Higher risk anal sex						
No	606	79.7	453	82.5	153	72.5
Yes	154	20.3	96	17.5	58	27.5
Alcohol						
No	434	57.0	295	53.7	139	65.3
Yes	328	43.0	254	46.3	74	34.7
Cannabis						
No	515	67.6	401	73.0	114	53.5
Yes	247	32.4	148	27.0	99	46.5
Erectile Drugs						
No	663	87.0	497	90.5	166	77.9
Yes	99	13.0	52	9.5	47	22.1
Poppers						
No	598	78.5	466	84.9	132	62.0
Yes	164	21.5	83	15.1	81	38.0
Ecstasy/MDMA						
No	723	94.9	524	95.5	199	93.4
Yes	39	5.1	25	4.6	14	6.6
GHB						
No	720	94.5	532	96.9	188	88.3
Yes	42	5.5	17	3.1	25	11.7
Crystal Methamphetamine						
No	685	89.9	520	94.7	165	77.5
Yes	77	10.1	29	5.3	48	22.5
Continuous Variables	Median	Q1, Q3	Median	Q1, Q3	Median	Q1, Q3
Male sex events number P6M	4	1, 15	4	2, 20	3.5	1, 12
Anal sex events number P6M	2	0, 8	2	0, 9	2	1, 8
Treatment optimism-skepticism scale	25	21, 28	24	20, 27	28	25, 32
Sexual altruism scale (communal)	3.5	2.8, 4	3.5	3, 4	3.2	2.5, 4
Sexual altruism scale (personal)	3.4	3, 3.9	3.6	3.1, 3.9	3.3	2.7, 3.9
Sexual sensation seeking scale	31	28, 34	30	28, 33	32	29, 35

Notes: P6M = Past six months; P12M = Past 12 months; PrEP = Pre-exposure prophlaxis; ART = Antiretroviral Therapy; AUDIT = Alcohol Use Disorders Identification Test; MDMA = 3,4-methylenedioxymethamphetamine; GHB = 3,4-methylenedioxymethamphetamine.

**Table 2 ijerph-18-03183-t002:** Univariable generalized estimating equations assessing higher risk anal sex among gbMSM in Metro Vancouver, 2012–2017.

	HIV-Negative/Unknown				HIV-Positive			
	Higher Risk Anal Sex (Yes 1457 vs. No 6664)				Higher Risk Anal Sex (Yes 1457 vs. No 6664)			
	OR	95% CI		*p*	OR	95% CI		*p*
Age								
16–29	Ref				Ref			
30–39	1.44	1.13	1.83	0.003	0.80	0.51	1.27	0.351
40+	1.39	0.98	1.97	0.069	0.56	0.35	0.88	0.013
Sexual Orientation								
Gay	Ref				Ref			
Bisexual	1.39	0.99	1.96	0.056	0.73	0.52	1.03	0.071
Other	1.02	0.72	1.44	0.915	1.06	0.76	1.46	0.736
Ethnicity								
White	Ref				Ref			
Asian	0.57	0.38	0.86	0.008	1.37	0.79	2.40	0.264
Indigenous	1.20	0.66	2.20	0.545	0.94	0.48	1.85	0.861
Latino	1.16	0.69	1.96	0.577	1.26	0.51	3.10	0.611
Other	0.89	0.45	1.77	0.738	3.05	1.85	5.04	<0.0001
Born in Canada								
No	Ref				Ref			
Yes	1.04	0.77	1.41	0.787	0.81	0.55	1.19	0.284
Neighborhood								
Downtown/West End	Ref				Ref			
Elsewhere Vancouver	0.72	0.57	0.90	0.004	1.26	0.95	1.67	0.106
Outside Vancouver	0.94	0.72	1.23	0.649	1.34	0.98	1.84	0.067
Formal Education								
Some or completed high school	Ref				Ref			
Any post-secondary training	0.78	0.56	1.09	0.144	1.51	0.97	2.35	0.066
Annual Income								
<$30,000	Ref				Ref			
at least $30,000	1.21	0.99	1.48	0.058	1.47	1.12	1.92	0.005
Current Regular Partner								
No	Ref				Ref			
Yes, but not common law/married	0.93	0.76	1.13	0.452	1.29	0.97	1.71	0.085
Yes, common law/married	0.87	0.67	1.12	0.274	1.19	0.92	1.55	0.185
Usage of PrEP								
No	Ref							
Yes	2.20	1.41	3.42	0.001				
Never heard of PrEP	0.81	0.66	0.98	0.033				
Escort Work								
No	Ref				Ref			
Yes, in P6M	1.77	1.18	2.65	0.006	1.17	0.72	1.90	0.531
Yes, not in P6M	1.72	1.20	2.48	0.004	1.29	0.88	1.90	0.193
Attended Group Sex P6M								
No	Ref				Ref			
Yes	1.44	1.21	1.73	<0.0001	1.15	0.91	1.45	0.238
AUDIT Zone								
Low-risk (scores 0 to 7)	Ref				Ref			
Medium-risk (scores 8 to 15)	1.01	0.84	1.21	0.935	1.02	0.73	1.43	0.903
Harmful (scores 16 to 19)	1.21	0.89	1.65	0.234	0.93	0.56	1.56	0.788
Possible dependence (scores 20 and over)	1.57	1.09	2.28	0.017	0.88	0.46	1.67	0.684
On ART Ever								
No					Ref			
Yes					2.65	0.35	19.90	0.343
Treatment Adherence P12M								
95% or greater					Ref			
<95%					1.15	0.90	1.48	0.266
Never on ART/start within 12 M					1.38	0.92	2.07	0.116
Viral Load <200								
No					Ref			
Yes					0.91	0.63	1.30	0.601
Prevention Strategies	OR	95% CI		*p*	OR	95% CI		*p*
Always Using Condoms								
No	Ref				Ref			
Yes	0.38	0.32	0.45	<0.0001	0.42	0.31	0.58	<0.0001
Seropositioning								
No	Ref				Ref			
Yes	1.69	1.39	2.06	<0.0001	1.77	1.41	2.23	<0.0001
No Anal Sex								
No	Ref				Ref			
Yes	0.76	0.65	0.90	0.001	0.77	0.63	0.95	0.013
Serosorting								
No	Ref				Ref			
Yes	1.18	1.00	1.39	0.046	1.09	0.89	1.35	0.403
Viral-Load Sorting								
No	Ref				Ref			
Yes	2.77	2.18	3.53	<0.0001	1.89	1.50	2.38	<0.0001
Withdrawal								
No	Ref				Ref			
Yes	1.08	0.90	1.30	0.410	1.44	1.14	1.81	0.002
Asking Status								
No	Ref				Ref			
Yes	1.08	0.90	1.30	0.410	0.98	0.81	1.19	0.860
Event-Level Variables	OR	95% CI		*p*	OR	95% CI		*p*
Alcohol								
No	Ref				Ref			
Yes	2.26	1.82	2.82	<0.0001	1.11	0.89	1.40	0.346
Cannabis								
No	Ref				Ref			
Yes	1.75	1.47	2.07	<0.0001	1.02	0.76	1.37	0.879
Erectile Drugs								
No	Ref				Ref			
Yes	1.67	1.25	2.23	0.001	1.03	0.83	1.29	0.765
Poppers								
No	Ref				Ref			
Yes	2.44	1.70	3.52	<0.0001	1.29	1.00	1.66	0.049
Ecstasy/MDMA								
No	Ref				Ref			
Yes	2.59	1.77	3.79	<0.0001	1.17	0.67	2.06	0.583
GHB								
No	Ref				Ref			
Yes	2.26	1.82	2.82	<0.0001	0.98	0.71	1.33	0.879
Crystal Methamphetamine								
No	Ref				Ref			
Yes	1.75	1.47	2.07	<0.0001	1.06	0.80	1.40	0.696
Continuous Variables	OR	95% CI		*p*	OR	95% CI		*p*
Male sex number P6M	1.00	1.00	1.01	0.001	1.00	1.00	1.01	0.267
Anal sex number P6M	1.01	1.00	1.02	0.001	1.00	1.00	1.01	0.038
Treatment optimism-skepticism scale	1.09	1.07	1.12	<0.0001	1.07	1.05	1.10	<0.0001
Sexual altruism scale (communal)	0.56	0.49	0.64	<0.0001	0.76	0.66	0.87	<0.0001
Sexual altruism scale (personal)	0.54	0.47	0.63	<0.0001	0.77	0.65	0.90	0.001
Sexual sensation seeking scale	1.12	1.09	1.16	<0.0001	1.09	1.05	1.13	<0.0001

Notes: P6M = Past six months; P12M = Past 12 months; PrEP = Pre-exposure prophlaxis; ART = Antiretroviral Therapy; AUDIT = Alcohol Use Disorders Identification Test; MDMA = 3,4-methylenedioxymethamphetamine; GHB = 3,4-methylenedioxymethamphetamine.

**Table 3 ijerph-18-03183-t003:** Univariable temporal trends and interactions of substance use and higher risk anal sex prevalence among gbMSM in Metro Vancouver, 2012–2017.

**Trend**	**HIV-Negative/Unknown**				**HIV-Positive**			
	**OR**	**95% CI**		***p***	**OR**	**95% CI**		***p***
Higher risk anal sex	1.006	1.002	1.011	0.009	1.006	1.001	1.012	0.025
Condomless anal sex	1.015	1.011	1.019	<0.0001	1.002	0.997	1.007	0.389
Knew neg vs. unknown	1.013	1.009	1.017	<0.0001	1.005	0.999	1.012	0.104
Knew pos vs. unknown	1.012	1.006	1.018	0.000	0.998	0.992	1.004	0.495
Alcohol	0.989	0.985	0.993	<0.0001	0.996	0.990	1.002	0.207
Cannabis	1.004	0.999	1.010	0.146	0.995	0.989	1.001	0.095
Erectile drugs	1.010	1.001	1.020	0.031	0.999	0.992	1.007	0.866
Poppers	0.998	0.990	1.005	0.561	0.991	0.984	0.998	0.007
Ecstasy/MDMA	1.012	1.000	1.024	0.053	0.997	0.976	1.018	0.760
GHB	1.000	0.986	1.015	0.952	1.001	0.990	1.012	0.869
Crystal methamphetamine	0.997	0.984	1.011	0.705	0.998	0.991	1.006	0.649
**Interactions**								
	**OR**	**95% CI**		***p***	**OR**	**95% CI**		***p***
Alcohol X trend	1.002	0.994	1.010	0.580	0.994	0.985	1.004	0.248
01/2012: User vs. not	1.051	0.778	1.419	0.746	1.372	0.942	1.997	0.099
12/2017: User vs. not	1.228	0.891	1.694	0.210	0.918	0.591	1.424	0.701
Cannabis X trend	1.004	0.995	1.013	0.410	1.004	0.995	1.014	0.358
01/2012: User vs. not	0.958	0.664	1.381	0.817	0.868	0.560	1.345	0.527
12/2017: User vs. not	1.267	0.855	1.878	0.239	1.188	0.761	1.856	0.448
Erectile drugs X trend	1.007	0.994	1.020	0.319	0.997	0.987	1.008	0.617
01/2012: User vs. not	1.797	1.100	2.936	0.019	1.145	0.715	1.833	0.574
12/2017: User vs. not	2.890	1.679	4.975	0.000	0.942	0.626	1.419	0.776
Poppers X trend	1.000	0.991	1.010	0.971	1.001	0.989	1.012	0.905
01/2012: User vs. not	1.737	1.254	2.404	0.001	1.258	0.797	1.987	0.325
12/2017: User vs. not	1.759	1.137	2.720	0.011	1.322	0.803	2.178	0.273
Ecstasy/MDMA X trend	1.012	0.993	1.031	0.226	0.981	0.962	1.000	0.054
01/2012: User vs. not	1.109	0.560	2.198	0.767	2.229	0.941	5.280	0.069
12/2017: User vs. not	2.543	1.171	5.523	0.018	0.568	0.250	1.287	0.175
GHB X trend	0.989	0.966	1.013	0.360	0.992	0.977	1.007	0.293
01/2012: User vs. not	3.506	1.680	7.317	0.001	1.315	0.639	2.705	0.457
12/2017: User vs. not	1.610	0.544	4.770	0.390	0.735	0.440	1.227	0.240
Crystal methamphetamine X trend	0.978	0.957	1.000	0.054	0.994	0.982	1.006	0.312
01/2012: User vs. not	5.048	2.376	10.726	<0.0001	1.324	0.903	1.941	0.150
12/2017: User vs. not	1.065	0.391	2.900	0.901	0.845	0.449	1.592	0.602

## Data Availability

Not Applicable.

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
