# Peer review of "Longitudinal Event-Level Sexual Risk and Substance Use among Gay, Bisexual, and Other Men Who Have Sex with Men"

_ijerph, 2021, doi:10.3390/ijerph18063183_

Round 1

Reviewer 1 Report

overall this is a well written and executed paper.  The design is appropriate and the analysis logical and appropriate given the research questions asked by the researchers.

There are however a few points of clarification required.  These are listed below.

  1. Substances such as methamphetamine, cannabis, amyl ni- 56 trate, cocaine, and erectile drugs.  Do you mean medically prescribed erectile drugs?  If yes, Why classify this under recreational drugs when it is a medically prescribed medication?

2.  The literature review includes a number of outdated references.  This needs to be addressed as the landscape is changing with prep and how this is now influencing sexual practices and negotiations in casual sex.  

3.  From point 2, in the conclusion can you discuss the implication of prep on future studies ... as the focus between condom and prep as prevention strategies may need rethinking.  How would your study be redesigned today? 

finally, are your results similar or different from those conducted outside of Canada?  

Reviewer 2 Report

The introduction is overall well-written. I do have a few comments that I believe will improve this section:

The literature review makes comments that are at times simplistic, particularly in regards to the cause-effect relationship between certain substance and sexual risk behaviour. At this stage It would be worth noting that there are differences in the patterns of drug use as well. Some drug types are particularly common among gay and bisexual men - particularly those associated with sex partying. This connection is not always clear in the introduction. Several studies have shown that men who do not or are less likely to use condoms are also more likely to participate in sexualised substance use where these behaviours are elevated. The cause-effect relationship is here not always clear even though chemsex has been linked to a meaningful number of HIV transmissions.

In a study of Spanish men (Gonzalez-Baeza et al., 2018) living with HIV, those engaged with chemsex were more likely to be diagnosed with other STIs, engage in ‘high-risk’ anal intercourse and reported a higher numbers of sexual partners. Another aspect that I am missing in both the introduction and the discussion is the potential effect of substance use on negotiating safer sex practices and the perception of high risk behaviours. I believe that these references are helpful for a stronger contextualisation of different aspects of the relationship between substance use and sex – including ‘high risk’ activities:

Bourne, A., et al., The Chemsex Study: drug use in sexual settings among gay and bisexualmen in Lambeth, Southwark & Lewisham. 2014, Sigma Research, London School of Hygiene & Tropical Medicine: London, United Kingdom.

González-Baeza, A., et al., Sexualized drug use (Chemsex) is associated with high-risk sexual behaviors and sexually transmitted infections in HIV-positive men who have sex with men: data from the U-SEX GESIDA 9416 study. AIDS patient care and STDs, 2018. 32(3): p. 112-118

I would also like to note that it might be useful to point out the positive consequences of substance use in the context of sex, pleasure from the heightened experiences provided by the drugs.

The methods section is well-written and clear. This is beyond the scope of this peer-review but I would like to mention that I have significant ethical concerns concerning the payment to participants for referring additional participants.

Beyond the points mentioned earlier, I enjoyed reading the discussion. Particularly comments regarding Poppers and PrEP were well-written and contextualised with the findings. There is now also evidence available demonstrating that the risk of Poppers use is indeed very limited:

Demant, D, et al., Harmless? A hierarchical analysis of poppers use correlates among young gay and bisexual men. Drug and Alcohol Review, 2018, 38: p. 465-472.

In regards to PrEP is might be worthwhile to comment on the changing nature of the perception of what constitutes high-risk behaviour.

Round 2

Reviewer 2 Report

The authors have addressed all comments. I have nothing to add.